# Contributions of Water-Related Building Installations to Urban Strategies for Mitigation and Adaptation to Face Climate Change

**Carla Pimentel-Rodrigues ***  **and Armando Silva-Afonso**

ANQIP (Portugal) and University of Aveiro/RISCO (Portugal), 3810-193 Aveiro, Portugal
* Correspondence: anqip@anqip.pt; Tel.: +351-234-092-597

**Abstract:** In addition to the possible contributions of buildings to mitigating $CO_2$ emissions, increased attention is being paid to the potential impacts of climate change on urban environments. According to the United Nations, about 54% of the planet's population currently lives in cities, but this percentage is expected to rise to 66% in 2050, which reveals the scale of this issue. This paper develops a reflection on the possible contributions of water-related building installations to mitigate emissions and increase urban area adaptation to the effects of climate change. One of the most promising solutions to facing climate change, which is analysed in detail in this paper, is combining rainwater harvesting systems with green roofs. However, in view of developing the necessary engineering projects, there are insufficient existing studies to estimate the parameters to be used in each location given their climate characteristics, particularly the monthly runoff coefficients, which constitute the key parameter for designing these installations in some regions. Some recent standards present generic theoretical values for designing these combined installations, but they are far from reality in some regions, such as the Mediterranean basin. Therefore, based on the data available in Portugal, this paper reports some of the results obtained from research on the values of the monthly runoff coefficients.

**Keywords:** climate change; adaptation and resilience; water efficiency in buildings; green roofs

## 1. Introduction

During the 21st century, climate change will continue under a range of possible greenhouse gas emission scenarios [1] and by the end of this century, the global average temperature will rise 2.6 to 4.8 °C from the present value, and sea levels will be 0.45 to 0.82 meters higher [2], significantly affecting coastal areas. More frequent and intense extreme weather events will result in a higher incidence of floods and droughts around the planet. Prolonged droughts will also reduce groundwater recharge and the subsequent impacts on water and sanitation services constitute a clear danger for development and health [3–5].

According to the United Nations, about 54% of the planet's population currently lives in cities, but this percentage is expected to rise to 66% in 2050, given that projections show that urbanization, combined with the overall growth of the world's population, could add another 2.5 billion people to urban populations by 2050 [1]. For this reason, climate change impacts in cities and buildings will be very significant and urban life will have to adapt and create resilience to more extreme weather conditions.

It is necessary to implement mitigation measures, consisting of interventions to reduce the sources of or enhance the sinking of greenhouse gases. At the same time, adjustments will be necessary to prevent or moderate the damage, thus increasing "resilience", that is, the capacity to manage harmful events, disturbances, or trends and respond so that buildings maintain their essential function, identity, and structure [2].

Water-related installations in buildings include water supply and drainage for rainwater and wastewater. They should contribute to adaptation to climatic changes and to adding more resilience to buildings [6], in addition to a possible contribution to mitigating the problem since an increase in water efficiency results in reduced energy consumption as a consequence of the water-energy nexus. It is very important to know what can be done to improve the traditional water-related systems and solutions or what needs to be done differently to ensure that buildings can handle the impacts of climate change in the future.

## 2. Climate Change Impacts and Response Strategies in the Urban Environment

### 2.1. Impacts on Temperature and Precipitation

An important impact of climate change that is expected to intensify in the next few decades is the increased intensity and frequency of heavy rainfall and other extreme weather events, such as heat waves [7]. Changes in precipitation are expected to differ from region to region, with some areas becoming more humid and others drier, increasing precipitation in high-latitude regions and decreasing it in most subtropical areas [8,9].

Regarding the European continent, southern and central Europe face increasingly more frequent heat waves, forest fires, and droughts. The Mediterranean area is also gradually becoming dry and, thus, even more vulnerable to droughts [10,11]. In fact, the Mediterranean basin is expected to be one of the regions most affected by climate change on the planet. Northern Europe is becoming increasingly wet and winter floods are likely to become more common. Being exposed to heat waves, floods, or a rising sea level, in the case of coastal cities, urban areas are often ill-equipped to adapt to climate change [12].

Changes in mean precipitation will impact groundwater recharge rates, which may affect the water supply [13,14], and, in semi-arid and arid areas, the salinization of shallow groundwater will intensify due to increased evaporation and water uptake by vegetation. With higher temperatures, there will be an increased demand for cooling (and hence power) in the summer and a decreased demand for heating in the winter. On the other hand, more frequent and intense winter rains lead to flooding in riverine areas and overloading public drainage systems [15–18].

According to online bulletins of the Copernicus Climate Change Service (C3S), implemented by the European Centre for Medium-Range Weather Forecasts on behalf of the European Union, in 2019, Europe was hit by two very intense heatwaves. Unprecedented temperatures broke records in many countries. When compared to the 2010–2018 median, large parts of Europe exceeded normal temperatures by over 20 °C in 2019. The effects were particularly strong on 25th July; however, temperatures of more than 15 °C above normal were felt throughout the whole period from 22nd July to 26th July.

The scientific community already envisages that similar events will be among the most significant threats that Europe will face in the foreseeable future. An increase in both the number and intensity of heatwaves is among the greatest threats facing Europe in the near future.

### 2.2. Mitigation Strategies

The primary mitigation strategies comprise carbon efficiency, technology energy efficiency, system and infrastructure efficiency, and service demand reduction through behavioral changes. Around the world, it is estimated that the building sector contributes as much as a fifth of the total global annual greenhouse gas emissions, making buildings the largest contributor to global greenhouse gas emissions, and this sector also consumes more than 32% of global final energy [2]. The major causes of this contribution are the extensive use of fossil fuel-based energy for thermal comfort, lighting, water heating, water supply and drainage, electrical equipment and appliances, and producing construction materials [19,20].

Given the massive growth in new construction, if nothing is done, greenhouse gas emissions from buildings will more than double in the next 20 years [20]. Considering a building's complete life cycle (construction, operation or use, and demolition), obtaining a significant reduction in greenhouse gases

emissions (GHG), mainly $CO_2$, will require effective measures to be taken during its use or operation phase, because this represents 80%–90% of the total energy consumed throughout its entire life cycle.

The use of green roofs on buildings, for example, can bring great advantages, not only in terms of mitigating energy requirements for thermal comfort, but also in terms of increased resilience, since they reduce the peak flow of surface water and increase the associated benefits of green infrastructure in urban areas. Hard engineering solutions will continue to play a role in adapting to climate change, and will also improve forecasting and preparedness, along with risk avoidance through planning controls.

Considering the water-energy nexus, reducing water consumption in the building cycle also produces significant energy efficiency. This is a result of reducing the energy needs for domestic hot water, to pressurize water in buildings, and also in public systems, such as pumping and the treatment of water and wastewater. Therefore, the nexus between water efficiency and energy efficiency should be one of the most important aspects when considering the contribution of buildings to mitigation strategies [21–23].

A study developed in a medium-sized city in Portugal (Aveiro) by ANQIP—a Portuguese association that works on water efficiency in buildings—found that energy savings due to using efficient products (classified as ANQIP labeling category "A" for product water efficiency) [24–26] lead to a reduction in emissions higher than 100 kg of $CO_2$ per capita, per year, compared to the present scenario. That value was obtained considering only heating domestic hot water in buildings and energy consumption in public networks. It should be noted that in Portugal, energy consumption for heating domestic hot water represents over 30% of the total housing energy consumption [26].

Taking as a reference the results of the study carried out in Aveiro, in Table 1, we summarize the savings obtained per component of the urban water cycle, per person and per household, considering an average value in Portugal of 2.3 persons per house and $CO_2$ current emissions of 269 g/kWh (according to online information of the main Portuguese energy market operator—EDP). It is thought that these results can be extrapolated to other urban contexts and may even be more relevant in cities with high-rise buildings since, in these cases, the pressurization needs are significant.

**Table 1.** Estimated energy savings and $CO_2$ reductions with the use of water-efficient products in buildings.

| Component of the Urban Water Cycle | Annual Energy Savings and $CO_2$ Reductions with the Use of Water-Efficient Products | | | | |
|---|---|---|---|---|---|
| | Per Person (kWh) | Per Person (kg of $CO_2$) | Per Family (kWh) | Per Family (kg of $CO_2$) | Percentage of the Total (%) |
| Building system (only sanitary hot water heating) | 368 | 99.0 | 846 | 228.4 | 87.0 |
| Public system of water supply | 32 | 8.6 | 74 | 19.9 | 13.0 |
| Public system of drainage and treatment of wastewater | 23 | 6.2 | 53 | 14.3 | |
| TOTAL | 423 | 113.8 | 973 | 262.6 | 100.0 |

This study shows the great importance of water efficiency measures in buildings as a contribution to reducing energy consumption in urban areas and mitigating GHG emissions. Naturally, these results may vary significantly with the characteristics of the water supply and drainage systems, especially with regard to public systems, but it should be noted that the system in the Aveiro region presents one of the lowest public water system energy consumptions in Portugal, due to its characteristics (flat city, superficial abstractions, etc.).

There are other studies that have been carried out about the relationship between water efficiency and energy efficiency in buildings [27–29] which reinforce the results of the above-mentioned study. However, it should be noted that the contribution of water efficiency in buildings to the reduction of energy consumption in public systems is generally neglected, but the study carried out in Aveiro showed that it can have a non-negligible value (around 13% in the case study).

In Aveiro, energy consumption in the public networks was 1.98 kWh/m$^3$ (1.16 kWh/m$^3$ in the public water supply system, and 0.82 kWh/m$^3$ in the public system of drainage and treatment of wastewater) [26]. Based on these values, it is possible to draw some conclusions about the energy consumed by some water uses in buildings, even in situations where only cold water is used, and which are not obvious to consumers. For example, it allowed us to determine that the discharge of a 6-litre flushing cistern implied an energy consumption close to 12 Wh in the public water supply and drainage networks, equivalent to a common 3W LED lamp connected for 4 h.

In general, the reuse of greywater and rainwater harvesting systems can also contribute to reducing the energy consumption. Indeed, as these systems reduce drinking water consumption in houses, they also reduce water flows and energy consumption in public networks.

For example, based on the value of 1.16 kWh/m$^3$ and on an estimate of the daily consumption in flushing cisterns of 145.8 liters per house in Aveiro, we can conclude that the use of rainwater in toilets, as an alternative to the use of drinking water from the main water system, would allow for an energy saving in the public water supply network of about 62 kWh per house and per year. Although rainwater harvesting systems may demand a pressurization system in the building, the corresponding energy consumption is equal to or less than that which occurs when the supply derives from the public network.

Compact installations for the direct reuse of greywater (toilet and washbasin combined, for example), reduce water consumption in buildings and also lead to saving water and energy. With regard to large installations for greywater reuse, when a centralized "conventional" treatment is employed to regenerate these effluents, we find that the energy consumed in the treatment makes the system "neutral" from an energy standpoint, i.e., the energy expended in treating greywater, about 1.8 kWh/m$^3$, is close to the energy saved in the urban water cycle. However, since the temperature of greywater from showers, for example, is generally above 30 °C, utilizing this thermal energy for pre-heating hot water will allow a saving of about 3 kWh/m$^3$, making these installations advantageous not only from the point of view of saving drinking water, but also from an energy standpoint.

In urban areas, reducing leaks in public water supply networks is also a well-known method to increase the water efficiency, which has additional economic advantages evident to water authorities. However, this measure is considered outside the scope of this paper, which focuses on buildings.

### 2.3. Processes of Adaptation and Increased Resilience

Buildings face a great risk of damage from the projected impacts of climate change and have already experienced a substantial increase in extreme weather damage in recent decades. More than half the urban areas projected for developing countries by 2030 have yet to be built, offering great potential for integrated adaptation planning, but special attention should also be paid to existing buildings. Furthermore, it would be of great interest to encourage good practice by incorporating climate change responses within engineering standards.

It is extremely important to develop suitable construction and weather-sensitive planning projects to promote the design of buildings and public spaces that are capable of dealing with the effects of climate change without significant damage. Using green roofs on buildings, for example, can bring great advantages, since they reduce the flow of surface water and increase green infrastructure and all of its associated benefits.

In addition to the contribution that buildings can make to mitigate the impacts of climate change on urban areas, for example, reducing flood peaks through green roofs and gardens [27] or reducing energy needs through water efficiency, it is important to specify the role of the water supply and drainage systems in buildings in relation to the resilience and adaptation of the building itself.

Constructing green roofs combined with rainwater harvesting systems in buildings can boost the advantages of each of these technologies [28,29]; their combination should also be considered a very promising solution to face climate change and increase sustainability in cities [30,31]. When designing a rainwater harvesting system combined with a green roof structure, several factors should be considered, such as the roof runoff coefficient [32].

## 3. Methodology

The runoff coefficient is a dimensionless parameter that represents the relationship between the total runoff volume from the roof and the total amount of precipitation in a certain time period [33]. In impervious roofs, where there is no loss of water by absorption and where water trapped or evaporated is not significant, it has a value near one. For a given reference period (usually day, month, or year), the multiplication of the runoff coefficient by the amount of precipitation in that period corresponds to the volume of water that can be used by the rainwater harvesting system.

In the case of green roofs, average annual values between 0.4 and 0.6 for extensive green roofs (green roofs with a maximum soil depth of about 150 mm), or between 0.1 and 0.4 for intensive green roofs (green roofs with a soil depth of 150 mm or more), are generally adopted in central Europe and the UK, according to the FLL (*Forschungsgesellschaft Landschaftsentwicklung Landschaftsbau*) guidelines [33].

In fact, these values depend on the characteristics of the roof, such as the type of plants used and the characteristics of the substrate, and are very dependent on the climatic conditions in the region, especially temperature and precipitation diagrams. In Mediterranean climates, monthly runoff coefficients are particularly important for sizing the storage tanks of rainwater harvesting systems [34] considering the existence of long dry periods, extending, in general, throughout the summer period. The monthly runoff coefficient represents the relationship between the total runoff volume from the green roof during a given month and the total amount of precipitation on the roof in that month added to the volume of possible watering done in that period.

From the perspective of integrating green roofs with rainwater harvesting systems, previous research has been conducted by the authors and other researchers on a conventional extensive green roof system in Oporto city (Portugal) [35]. This study revealed low runoff coefficient values, but allowed the development of an expression to predict the monthly runoff coefficients for this type of conventional green roof. The extensive pilot system adopted (Figures 1 and 2) followed the typical extensive green roof structure: geotextile membranes, a water holding capacity layer using expanded clay, and a 10 cm growing substrate composed of a mixture of expanded clay and organic matter. The pilot green roof was established with three different common aromatic plant species: *Satureja montana*, *Thymus caespititius*, and *Thymus pseudolanuginosus*.

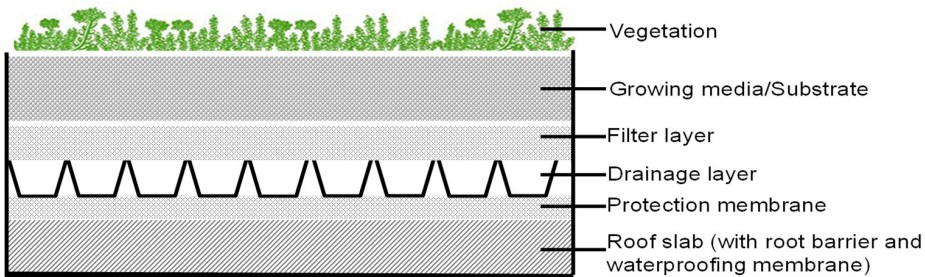

**Figure 1.** Green roof schematic representation [35].

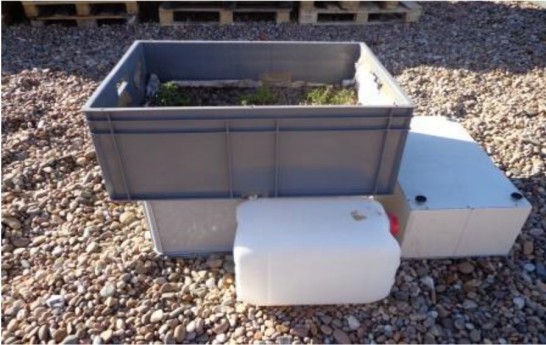

**Figure 2.** Green roof experimental set up [35].

Although the pilot system was relatively small ($0.70 \times 0.70$ m$^2$) and could lead to scale distortions in the results, it was thought that these errors would not be significant and that the results met the objective of obtaining a practical mathematical expression that would allow, with acceptable approximation, the determination of average values of the monthly runoff coefficient for a typical extensive green roof [35].

Measurements of the pilot green roof allowed the development of the following expression for monthly runoff coefficient prediction [35]:

$$C_M = \text{K} \frac{(P_M + R_M)}{(2T_M - T_{M-1})^{1.2}} \qquad (1)$$

where K = 0.016 ($^{\circ}$C$^{1.2}$ mm$^{-1}$), $C_M$ is the runoff coefficient of month $M$, $P_M$ is the precipitation of month $M$ (mm), $R_M$ is the watering in month $M$ (mm), $T_M$ is the mean air temperature during month $M$ ($^{\circ}$C), and $T_{M-1}$ is the mean air temperature during month $M-1$ ($^{\circ}$C).

The expression obtained, which significantly depends on the temperature in previous periods and precipitation, has similarities with the well-known Turc formula [36] that has been widely used in hydrological studies to determine flow deficit, which can be considered an indicator of its consistency [35]. It should be noted that, where the application of the formula leads to a $C_M$ value greater than 0.50, it is recommended that this value is adopted as a maximum, taking into account indications of the European Standard EN 16941-1 [37].

Assuming its validity in Portugal, Formula (1) was applied to 12 weather stations (Figures 3 and 4), where it is possible to obtain temperature and precipitation values, to calculate theoretical values for the monthly runoff coefficients for green roofs with characteristics similar to those of the pilot system [32–38] and find patterns related to the climatic nuances in the country. For these weather stations, the historical record of monthly rainfall and temperatures is available on the Portuguese government's SNIRH – Sistema Nacional de Informação de Recursos Hídricos website (https://snirh.apambiente.pt/).

Most of Portugal has a Mediterranean climate, but northern Portugal has a significant Atlantic influence, where the Mediterranean climate is less dominant. This is the case for stations 2, 7, and 10. Station 5, although also located in the north of Portugal, is situated in the Douro valley, which has a very specific local climate. The long mountain range on the west rim of this region protects the valley from the cold and humid winds that sweep in from the Atlantic, creating drier conditions than along the coast, but also colder winters and warmer summers. It is due to its special climatic conditions that this valley has become the birth of one of the world's most well-known drinks, Port Wine.

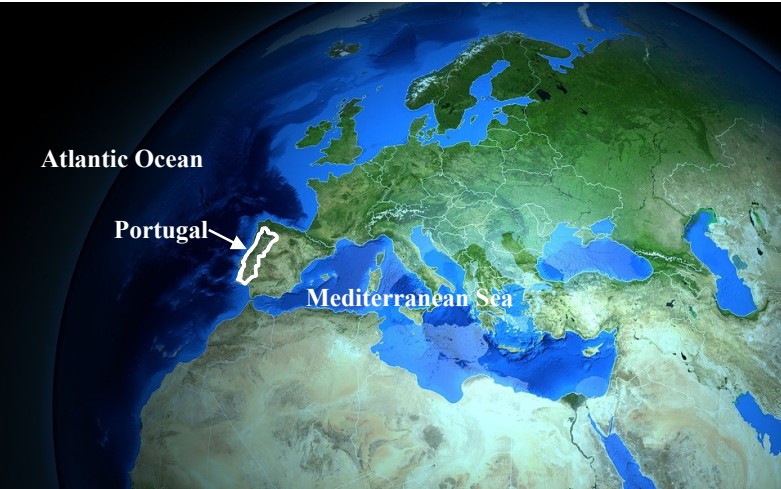

**Figure 3.** Location of Portugal close to the Atlantic Ocean and the Mediterranean Sea.

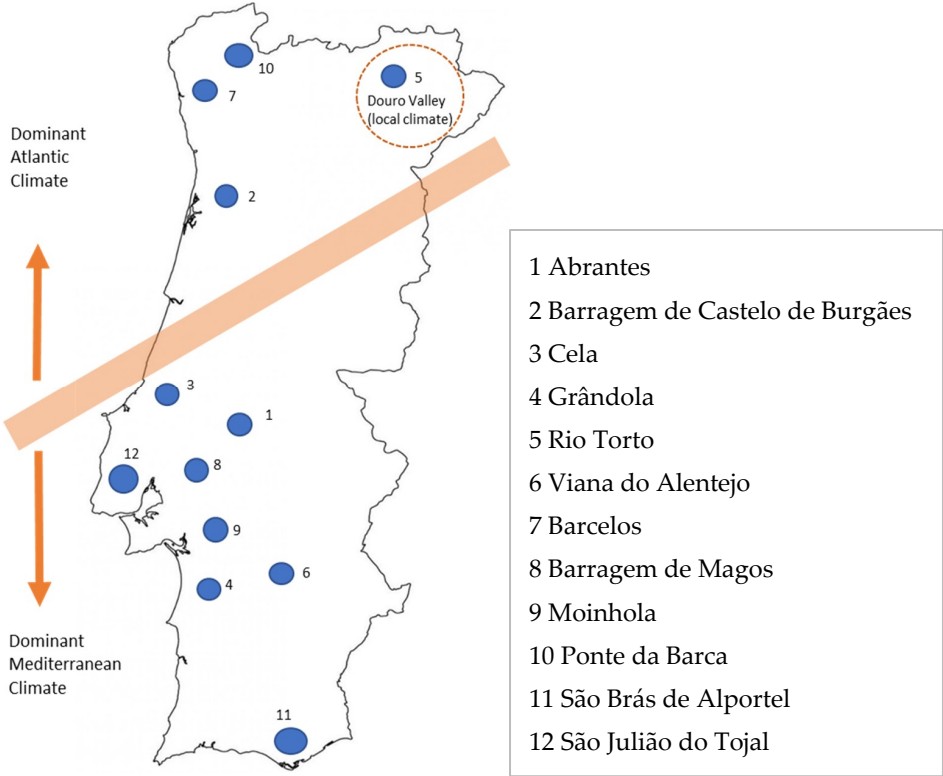

**Figure 4.** Portuguese weather stations with information available.

## 4. Results and Discussion

In general, there are two climate change impacts that are directly linked to water-related building installations: the increased intensity of heavy rainfall and extreme heat waves. Table 2 summarizes the main measures to be adopted in buildings with a view to promoting the adaptation and resilience of buildings against these impacts.

**Table 2.** Measures to be adopted for promoting the adaptation and resilience of buildings against climate change impacts.

| Type of Climate Change Impact. | Measures to be Adopted in Water-Related Installations to Promote an Increased Resilience of the Building | |
| --- | --- | --- |
| | **New Buildings** | **Existing Buildings** |
| Increased heavy rainfall intensity | - Review design standards by integrating new weather data or higher safety coefficients;<br>- Construct green roofs (preferably mandatory);<br>- Install rainwater harvesting systems (preferably mandatory). | - Review rainwater drainage pipe sizing, especially stacks and drains (in gravity systems), and analyse the need for new emergency overflow outlets (specifically in siphon systems);<br>- Install rainwater harvesting systems (if possible). |
| Extreme heat waves (water scarcity) | - Review design standards considering greater capacity in water tanks (when they exist in the building);<br>- Install rainwater harvesting systems and/or greywater reuse systems;<br>- Apply water-efficient products (preferably mandatory). | - Conduct water efficiency audits;<br>- Install rainwater harvesting systems and/or greywater reuse systems (if possible);<br>- Exchange installed devices for more efficient ones or apply flow or volume reducers. |

In the case of increased heavy rainfall intensity, it is necessary to adjust the design standards for new buildings and review the design of rainwater drainage in existing buildings. The latter aspect is more delicate with regard to rainwater siphonic systems [39], whose capacity to respond to unforeseen flow increases is smaller since it is known that, in these systems, a flow slightly above the design flow is sufficient to cause a rapid increase in the depth of water on the roof. Placing more emergency overflow outlets could be the solution. Reviewing the design standards should include new weather data and/or higher safety coefficients.

Regarding extreme heat waves and the inherent risk of water scarcity, adjusting standards is again necessary, especially with regard to reviewing water tank sizing and increasing the efficiency of water

use in buildings. Rainwater harvesting and greywater reuse should be promoted, with this being the first solution particularly suited to answer the many impacts of climate change because it simultaneously reduces the flood peaks in urban areas and promotes additional water storage in buildings.

The combination of green roofs and rainwater harvesting systems in buildings seems to be a very promising constructive solution, but this technology requires a refined knowledge of runoff coefficients, which show great variability due to roof characteristics and local climatic conditions. It is therefore important to determine reference values in each specific climate region for the design of these combined solutions, improving knowledge in this field that currently boils down to indicative values in some standards, and which research has shown to be non-generalizable.

Following on from previous research, which led to a formula for determining monthly runoff coefficients ($C_M$) in Portugal, applicable to extensive green roofs, we sought to analyse the variability of their values across the territory. Given that Portugal, despite its small size, is a country with two very characteristic types of climate (Mediterranean in the south and Atlantic in the north), we also sought to find characteristic patterns in the $C_M$ values for these two types of climate.

The results obtained with the application of the formula to the average values calculated from the records available for hydrological years 1980/1981 through 2017/2018, are shown in Figure 5. The monthly runoff coefficient values correspond to a minimum of zero and a maximum of 0.46. It should be noted that these values are clearly lower than the average annual runoff coefficients proposed in the literature for green roofs in central/northern European countries (0.5 for extensive green roofs and 0.3 for intensive green roofs) [37,40].

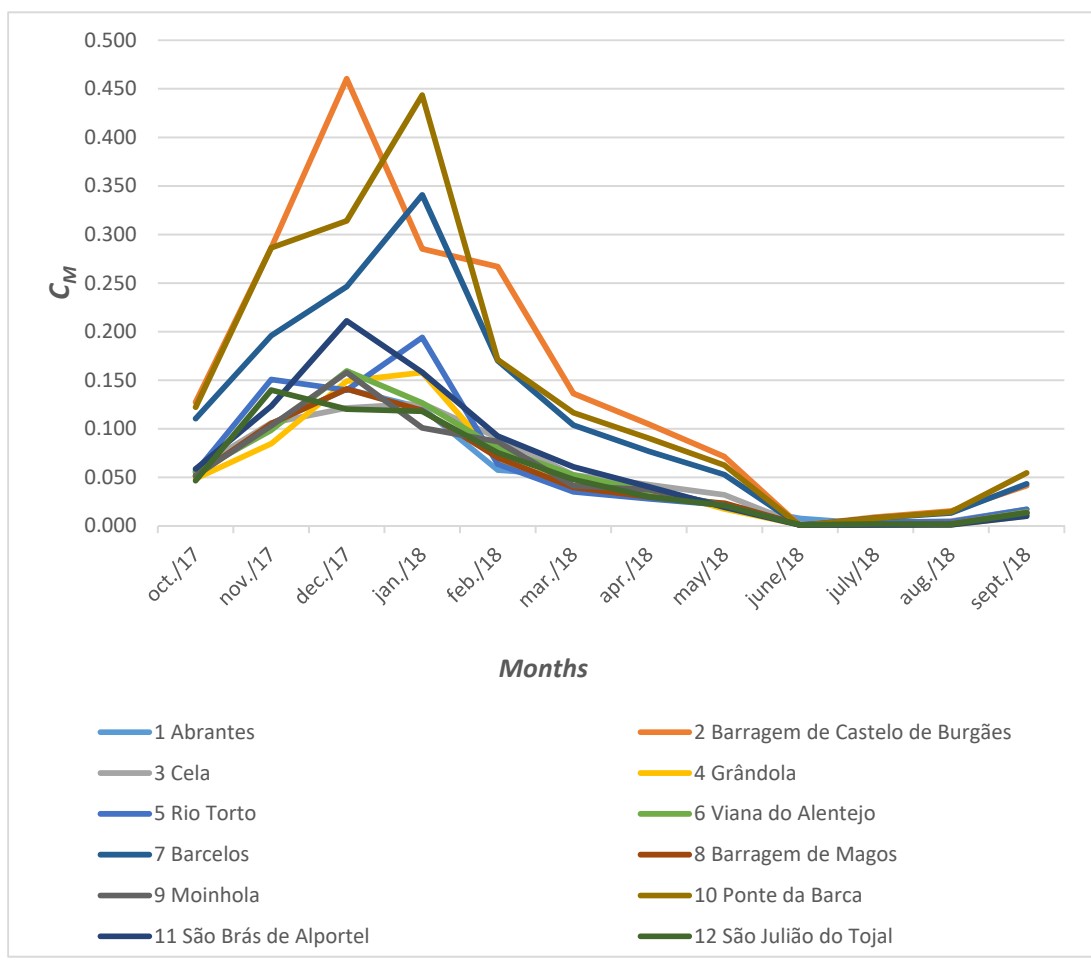

**Figure 5.** Monthly runoff coefficient for the different stations (2017–2018).

Interestingly, in the three weather stations with an Atlantic climatic influence, the observed results are significantly different from the others, following standards close to those recommended by FLL. On the other hand, it is notable that the results follow a very different, but internally consistent, pattern in the weather stations with dominant Mediterranean climates.

An obvious conclusion is that the green roofs combined with rainwater harvesting systems in Mediterranean climates will require larger catchment areas and mainly higher volumes in the storage tanks for technical feasibility. This requirement will, of course, result in higher costs for these combined solutions precisely in the regions where they will be most interesting or even necessary.

## 5. Conclusions

Dealing with climate change is one of the major challenges facing mankind in the 21st century. It is necessary to simultaneously implement mitigation measures consisting of interventions to reduce the sources or enhance the sinking of greenhouse gases and adjustments to the "new" climate and its effects. This intervention essentially seeks to prevent or moderate the damage, which are known as processes of adaptation and increasing resilience.

Buildings play an essential role in these processes, not only in relation to mitigation measures, but also in the need to be adapted and acquire a higher resilience. Building installations for water supply and drainage make specific contributions in all these processes and can significantly contribute to mitigation; they are also essential to adaptation and increasing resilience in the face of some of the projected impacts of climate change.

Increasing the water efficiency in buildings should be considered a priority measure, but some solutions—such as green roofs or rainwater harvesting systems in buildings—can also greatly contribute to a very appropriate response to the impacts of climate change. These solutions should be widely generalized, possibly through a mandate in some regions. However, the design of these systems greatly depends on the green roof characteristics and the particularities of local or regional climates, so further research in this field is needed.

In Mediterranean countries, which are among those most affected by climate change, the cost of these solutions may be significantly higher, although their technical and economic viability may also remain interesting given the growing scarcity of the resource and the need for adaptation and increased resilience to climate change effects and mitigation measures.

Previous studies carried out for a conventional extensive green roof pilot system in Oporto city, Portugal, allowed the development of a practical expression to predict a 'monthly runoff coefficient' for a typical extensive green roof, which is the parameter usually used for sizing storage tanks in Mediterranean countries in engineering projects. It should be noted that, in terms of future work, this study will need to be continued with more extensive models or even in real cases, as the small size of the pilot used may imply distortions by a scale effect, which should be evaluated in order to perfect the mathematical expression obtained.

To evaluate the effect of local climatic conditions on the monthly runoff coefficients, this expression was applied in different regions of Portugal, a country that has a Mediterranean climate in the centre and south, but a temperate Atlantic climate in the north, similar to the central European climate. The theoretical results, not considering changes in the characteristics of green roofs, show wide variations in monthly runoff coefficients, with a minimum of zero and a maximum of 0.44. The results show that these coefficients significantly depend on the climatic characteristics of the site, indicating very relevant differences between the areas of Portugal where an Atlantic climate is dominant and the predominantly Mediterranean climate zones, but also that they are consistent within each of these specific local areas.

The consistency of the expression (1) and results support the conclusion that it can be used in other countries or regions with similar climatic characteristics (southern Europe with a Mediterranean climate and central Europe with an Atlantic climate), providing much stricter calculation values than those indicated in the current standards. However, in view of a broader generalization of the study and the application of the present methodology to other countries or regions with different climate patterns,

it will be necessary to carry out an initial assessment of the value of the coefficient K of the mathematical expression and the value of the exponent of the denominator, via a local experimental study.

**Author Contributions:** The authors made similar contributions to the development of the article. Conceptualization, A.S.-A. and C.P.-R.; Methodology, A.S.-A. and C.P.-R.; Validation, C.P.-R.; Formal analysis, A.S.-A.; Investigation, A.S.-A. and C.P.-R.; Writing—original draft preparation, C.P.-R.; Writing—review and editing, A.S.-A.; Supervision, A.S.-A.; Project administration, C.P.-R.; Funding acquisition, C.P.-R.

**Funding:** The authors are grateful for the financial support provided by the Portuguese Foundation for Science and Technology (FCT) under Project (POCI-01-0145-FEDER-016852), co-funded by the Operational Program for Competitiveness and Internationalization (POCI) of Portugal 2020 with the support of the European Regional Development Fund (FEDER).

**Conflicts of Interest:** The authors declare no conflict of interest.

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
