# Peer review of "Contributions of Water-Related Building Installations to Urban Strategies for Mitigation and Adaptation to Face Climate Change"

_applsci, doi:10.3390/app9173575_

Round 1

Reviewer 1 Report

This manuscript is very interesting and it deals with a very important and a timely topic. The study has increased our understanding in the area of mitigating the effect of climate change on urban areas now and in the future through the use of rainwater harvesting and reducing energy consumption on a world that is becoming more urbanized.  Having said this, there are few points that need to be addressed before the manuscript can finally be accepted for publication in Applied Sciences. The theoretical and conceptual framework is very good; however, some of the applied part of the manuscript needs more explanation and organization. The points that need to be addressed are:

There has to be a separate section that deals with the Study area. Since the manuscript is dealing with rainwater harvesting and heat waves or increase of temperature, the reader would like to know about records of precipitation and temperature. Obviously, the two maps in page 7 should be included in this section. Another small section about the methodology is needed in which the authors give us more explanation about the data collection and analysis. Although some of it has been mentioned in the manuscript. On page 5, the authors have mentioned in the paragraph before the last that rainwater harvesting systems reduces ………. and energy consumption in public network. Would it be possible to give example of the way or how it can reduce energy consumption in public network.

Author Response

The authors would like to thank the reviewers for their contributions and suggestions for improving the text.

As proposed, additional information on heat waves and temperature records was included. A new section with the methodology has also been included. Also included is an example of possible reduction of energy consumption in public networks of water supply resulting from the installation of rainwater harvesting systems in buildings.

Reviewer 2 Report

Dear authors,

Thank you for this concise piece of research. There are a few comments I would like to pose:

- further explain both of the experiments conducted in detail, and present the results

- in line 150 to 152; the explanation of the runoff coefficient needs to be clarified or reworded

- lines 240 onward; further explanation on the climate features, or at least for the highlighted ones (Douro valley) is needed.

- in line 260; the figure contains a symbol which is not explained. Please do.

- in the bibliographic references; Please review the use of capital letters other than for acronyms.

Please consider my comments as recommendations, not binding.

with kind regards,

the reviewer

Author Response

The authors would like to thank the reviewers for their contributions and suggestions for improving the text.

The authors reorganized the paper, trying to clarify the work done and include new subsections. As regards the runoff coefficient, clarification was also made. An explanation of the specific climate of the Douro Valley was also included in the text. Figures 3 and 4 have been improved and bibliographic references have been revised, although conditioned by the requirements of the editor's template.

Reviewer 3 Report

 see attachment 

Author Response

The authors would like to thank the reviewers for their contributions and suggestions for improving the text.

The questions raised by the reviewer are pertinent, but most have already been answered in a previous paper and the authors considered that the repetition of these points in this paper was not justified. In fact, the questions concerning the pilot system have already been presented, discussed and accepted by the scientific community in the biographical reference [32] (Monteiro, C., Calheiros, C., Pimentel-Rodrigues, C., Silva-Afonso, A., Castro, P., Contributions to the design of rainwater harvesting systems in buildings with green roofs in a Mediterranean climate. Water Science and Technology, 2016, 73.8, 1842-1847). Regarding the remaining issues, the paper was reorganized, the figures were corrected and the institution providing the weather data in Portugal is indicated.

Round 2

Reviewer 3 Report

The manuscript has been now improved. Nevertheless, the scientific contribution of the paper to the existing literature is still not well pointed out. The authors should suggest how to generalize the proposed methodology in order to be applied to other cases of study. A discussion about the error related to the scale distortion due to the dimensions of the pilot system has not been included. In my opinion, this part of the study still requires some more comments.

Author Response

The authors would like to thank reviewer 3 for the new contributions and suggestions for improving the text.

In this new version the authors welcomed and tried to adopt these new observations and suggestions. Responding specifically to the new observations of the reviewer 3, we refer to the following:

Reviewer 3 - Regarding the monthly runoff coefficients, the authors have included new paragraphs with the methodologies proposed to generalize this study were included in the conclusions.

The authors fully agree with the reviewer on the issue of possible distortion errors resulting from the pilot scale. However, given that this issue was dealt with in an earlier paper published in another journal, involving the participation of another researcher as the first author [32], they consider that it was not appropriate to use this paper to reopen this discussion. However, given the importance of the issue, which is recognized, a paragraph was also added recommending the assessment of possible distortions in future work.
